DOI: 10.1038/s41467-018-04886-2　　OPEN

# A molecular neuromorphic network device consisting of single-walled carbon nanotubes complexed with polyoxometalate

Hirofumi Tanaka [1], Megumi Akai-Kasaya[2], Amin TermehYousefi[1], Liu Hong[3,5], Lingxiang Fu[1], Hakaru Tamukoh[1], Daisuke Tanaka[3,6], Tetsuya Asai[4] & Takuji Ogawa[3]

In contrast to AI hardware, neuromorphic hardware is based on neuroscience, wherein constructing both spiking neurons and their dense and complex networks is essential to obtain intelligent abilities. However, the integration density of present neuromorphic devices is much less than that of human brains. In this report, we present molecular neuromorphic devices, composed of a dynamic and extremely dense network of single-walled carbon nanotubes (SWNTs) complexed with polyoxometalate (POM). We show experimentally that the SWNT/POM network generates spontaneous spikes and noise. We propose electron-cascading models of the network consisting of heterogeneous molecular junctions that yields results in good agreement with the experimental results. Rudimentary learning ability of the network is illustrated by introducing reservoir computing, which utilises spiking dynamics and a certain degree of network complexity. These results indicate the possibility that complex functional networks can be constructed using molecular devices, and contribute to the development of neuromorphic devices.

[1] Graduate School of Life Science and Systems Engineering, Kyushu Institute of Technology (Kyutech), 2-4 Hibikino, Wakamatsu, Kitakyushu 808-0196, Japan. [2] Graduate School of Engineering, Osaka University, 2-1 Yamadaoka, Suita, Osaka 565-0871, Japan. [3] Graduate School of Science, Osaka University, 1-1 Machikaneyama, Toyonaka, Osaka 560-0043, Japan. [4] Graduate School of Information Science and Technology, Hokkaido University, Kita 14, Nishi 9, Kita-ku, Sapporo 060-0814, Japan. [5] Present address: School of Chemical and Material Engineering, Jiangnan University, No. 1800 Lihu Avenue, Wuxi 214112, China. [6] Present address: School of Science and Technology, Kwansei Gakuin University, 2-1 Gakuen, Sanda, Hyogo 669-1337, Japan. Correspondence and requests for materials should be addressed to H.T. (email: tanaka@brain.kyutech.ac.jp)

Brain-inspired computing has attracted considerable attention in recent years for its potential to perform intelligent, robust and low-power computations in situations in which conventional algorithm-based computing on Neumann-based computers may falter[1]. Neumann- and silicon-based AI accelerators, such as MIT and NVIDIA's Eyeriss processors[2], a machine-learning supercomputer (DaDianNao)[3] and commercial general-purpose graphic processing units (GP-GPUs), are used to create intelligent artefacts with learning and cognitive abilities. The silicon-based neural accelerators mentioned above provide such intelligent functions; however, they are based on advanced computer science and engineering, and are not based on contemporary neuroscience, which causes their applications to be limited to those such as pattern classification and inference. Neuromorphic hardware, on the other hand, is based on neuroscience, and provides excellent opportunities to replicate higher-level brain functions. In contemporary neuromorphic hardware (e.g. IBM's neurosynaptic chip (TrueNorth)[4], analogue or digital neuromorphic integrated circuits[5,6], etc.), artificial spiking neurons that mimic nerve impulse (spike) generation and the construction of their dense and complex networks are essential. Coding neuronal information using spikes is functionally important upon transmitting actions on neuronal membranes (active transmission lines) in noisy and unreliable environments[7]. The usefulness of spiking neural networks in practical applications has not become completely clear; however, it has recently been demonstrated that complex and spontaneous dynamics generated by large-scale spiking neural networks are useful for blind source separation[8], reservoir computing[9] and so on.

In present neuromorphic systems, both the integration density and wiring complexity, which directly represent the potential intelligent information processing ability, are much lower than those of human brains[10], because present major neuromorphic hardware is only composed of silicon complementary metal–oxide semiconductor (CMOS) devices. In this report, we present an extremely dense, molecular neuromorphic network device, composed of a network of single-walled carbon nanotubes (SWNTs) complexed with polyoxometalate (POM)[11] as an alternative to the present silicon CMOS analogue and digital neural processors.

At least two types of devices are required to construct analogue neuromorphic hardware: synaptic devices and neuronal membranes. The synaptic device lies at the intersection between the axonal and dendritic wires of the neuron devices and acts as a memristive junction whose coupling strength is stored. A synaptic device consisting of a network of carbon nanotubes (CNTs) has been proposed[12]. The neuronal membrane (neuron) device emits spikes (nerve impulses) and transmits the impulses to other neurons via axonal and dendritic wires[13]. Such neural membrane

devices have not been further explored in spite of recent significant advances in materials science. SWNTs are good candidate materials for neural membrane devices because metallic CNT-based conductors generate large electrical noise with rich dynamics[14–16]. Moreover, it has been observed that the electronic state and conduction mode of an SWNT vary enormously with the type of molecular species adsorbed[17,18]. Phosphododecamolybdic acid, ($H_3PMo_{12}O_{40}$; $PMo_{12}$ hereafter), is one of the POMs exhibiting reversible multi-electron redox properties[19–21], electronic versatility[22,23] and negative differential resistance (NDR) on highly ordered pyrolytic graphite (HOPG)[24].

In this report, we present a complexed SWNT/POM network molecular neuromorphic device consisting of a dense and complex network of spiking molecules ($PMo_{12}$ particles[25]) that imitates a large-scale spiking neural network. Based on experimental studies of the fabricated SWNT/POM network, we first discuss its NDR and noise properties and then demonstrate its collective impulse generation. To reveal the possible impulse generation mechanism, we propose an abstract model of the network by assuming a two-dimensional (2D) structure of molecular junctions and demonstrate that the model yields results in good agreement with the experimental results. In Supplementary Note 2, we illustrate the potential of the SWNT/POM network model for neuromorphic reservoir computing[26,27] by demonstrating basic learning ability of the network.

## Results

**Neuromorphic SWNT/POM device and experiments**. The typical structure of an individual SWNT/POM complex prepared on a Si substrate observed by atomic force microscopy (AFM) is shown in Fig. 1a. Both the bundled SWNTs and adsorbed POM particles have diameters of a few nanometres. The total thickness of the complex is around 10 nm. The current-voltage (I–V) characteristics measured on the individual complexes by point-contacted current-imaging (PCI-) AFM[18,28,29], are presented in Fig. 1b. Because the voltage sweeping speed was fixed and rapid in the AFM controller, this measurement was performed purely as a preliminary check. Several peaks are apparent; i.e. the current increases non-monotonically as the bias voltage increases due to the NDR characteristics of the SWNT/POM device. The NDR observed for the POM particle is considered to be a result of multi-redox activities within the charge container. The NDR peak position is closely related to the reduction potential as well as the electronegativity of the counter-action[23]. However, the NDR positions in the I–V plots shown in Fig. 1b are not unique. Some reactions with surrounding counter-ions might have substantial influence on the NDR characteristics; however, the physics of the NDR phenomena of POM particles has not yet been fully understood.

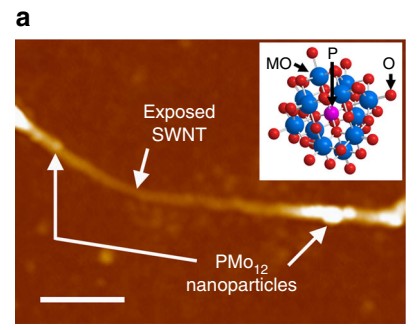

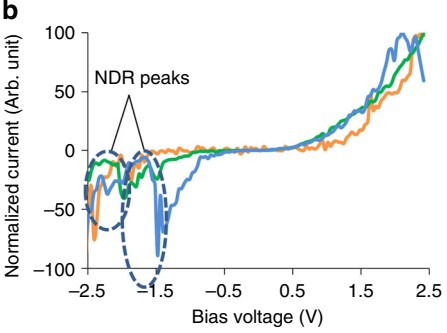

**Fig. 1** SWNTs complexed with POM. **a** AFM image of the SWNT/POM complex. (Inset) Molecular structure of $PMo_{12}$, reproduced from its reported X-ray crystallographic structure (scale bar = 200 nm). **b** I–V characteristics of the SWNT/POM complex, measured by PCI-AFM, where several NDR-related peaks are observable

We studied the electrical properties of the SWNT/POM network. Figure 2a illustrates the assumed network structure where the yellow cuboids, black tubes and purple spheres represent the terminal electrodes, SWNTs and POM particles, respectively. Figure 2b represents a micrograph of the fabricated SWNT/POM network device having multiple terminal electrodes, denoted (1–6), with different gap distances between the electrodes. Two kinds of samples, namely 'sample A' rinsed by ethanol and 'sample B' rinsed by DI water, were evaluated.

Electrical measurement results for sample A are shown in Figs. 2c and 3a, b, e, f. Figure 2c presents sampled current plots for sample A over time, representing the distribution of the current magnitudes with stepwise-increasing $V_B$ (from 0 to 125 V). Gaussian-like noises were generated, and the mean and variance of each distribution were increased as the bias voltage $V_B$ was increased. Figure 3a shows the $I$–$V$ characteristics of sample A, lying between electrodes (1) and (2) (Fig. 2b), which exhibit an NDR peak (marked with a red arrow) between $V_B$ of 125 and 150 V across the terminal electrodes. The $I$–$V$ characteristics were measured in air at room temperature with averaging over 100 power-line cycles (PLCs) with a 60-Hz power line in the environment. When $V_B$ was further increased to 150 V, the current became unstable and non-Gaussian distributions appeared, as shown in the inset of Fig. 2c, resulting in periodic/aperiodic current impulse generation. By applying a voltage higher than 150 V, the system became unstable, indicating current impulse generation, as shown in Fig. 3b. This condition allowed the generation of periodic current impulses, as shown in the inset of Fig. 3b.

The $I$–$V$ characteristics of sample B, lying between electrodes (1) and (2), are shown in Fig. 3c. When the absolute bias voltage across the terminal electrodes $|V_B|$ is larger than approximately 80 V, large hysteresis is observable between the forward and backward sweeps, exhibiting NDR characteristics, as well as unstable currents. The generation of unstable currents at lower $|V_B|$ (<80 V) was suppressed by accelerated ion transfer in the water-treated samples, as compared to that in the ethanol-treated samples such as sample A. The NDR peaks of the SWNT/POM network appeared for both negative and positive bias in Fig. 3c, whereas the peaks in Fig. 1b appear only for negative polarity. The polarity of the NDR appears to depend on the voltage sweep direction in Fig. 1b, because the sweeping speed is too fast to cause redox in the same polarity sweep. Even the NDR

originating from the redox of $PMo_{12}$ should be irrelevant to the polarity. Figure 3d shows plots of the current versus time for sample B when $V_B = 80$ V, where periodic/aperiodic current oscillations (at approximately 25 Hz) and random current impulses are observable. Note that such impulses can also be observed in a POM/SWNT complex device having a microscale channel under $V_B < 1$ V. See Supplementary Note 1 for the details.

Temporal sequences of inter-spike intervals (ISIs) of the impulse trains were measured to quantify the self-similarity in the impulse generation process. Figure 3e, f show the Poincaré plots[30] against the applied bias voltages $V_B$ and concentration ratio of POMs to SWNTs, respectively. As an example, short sequences of impulses and their ISIs are shown in the inset of Fig. 3e, where $t_n$ represents the $n$-th ISI ($n$ is the natural index number, and the maximum value is equal to the number of measured impulses minus 1). Each Poincaré plot was then created by plotting a dot at $(t_n, t_{n+1})$ for all $n$, to distinguish chaos from randomness visually. As shown in Fig. 3e, the Poincaré plots of the ISI sequences obtained from sample A for various $V_B$ do not exhibit any self-similarity, indicating that the generated impulse train was a random (not chaotic) sequence. This phenomenon was reproduced even when the concentration ratio of POMs to SWNTs $R_{P2S}$ was varied.

Furthermore, we found that the ISI distribution can be modified by changing $R_{P2S}$. In Fig. 3f obtained from sample A, ISI distributions for specific $R_{P2S}$ are represented. With increasing $R_{P2S}$, these interval distributions decrease in size and shift to the upper right, which represents the tendency for longer ISI impulse generation with higher $R_{P2S}$. In fact, increasing the amount of POM also increased the size of the POM nanoparticles, which directly affected the electrical properties of the POM junctions by increasing the capacitance, which explains why the impulse generation period also increased.

The experimental results indicate that impulse generation in the POM/SWNT network is closely related to the existence of NDRs in the $I$–$V$ characteristic of the device. Since similar NDR characteristics are also observable in single POM junctions, we hypothesise that conductance switching at a POM junction having multiple-redox ability is the origin of the impulse generation (the term 'multiple-redox' reflects the fact that a POM molecule is capable of having multiple charges). Electronic chemical measurements have proven that $PMo_{12}$ can store up to 24 electrons entailing the molecular structural change; hence, the $PMo_{12}$ is called an 'electron sponge'[31]. On the other hand, it is

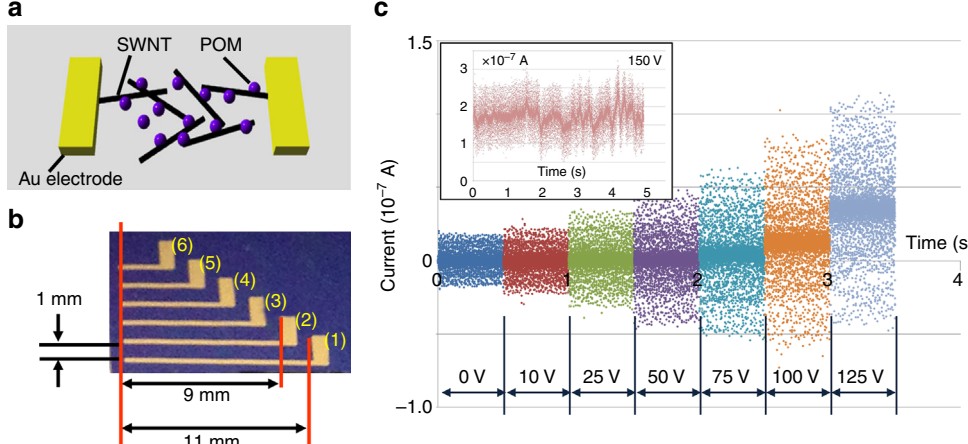

**Fig. 2** Experimental setup and noise generation of the SWNT/POM network. **a** Schematic of a network with the SWNT/POM complex network. The yellow cuboids, black tubes and purple spheres represent the terminal electrodes, SWNTs and POM particles, respectively. **b** Photograph of the substrate including six terminal electrodes. The entire substrate was covered with the SWNT/POM complex. **c** Sampled current density over time, representing the current magnitude distributions, with the bias voltage $V_B$ increasing stepwise across the electrodes from 0 to 125 V for sample A

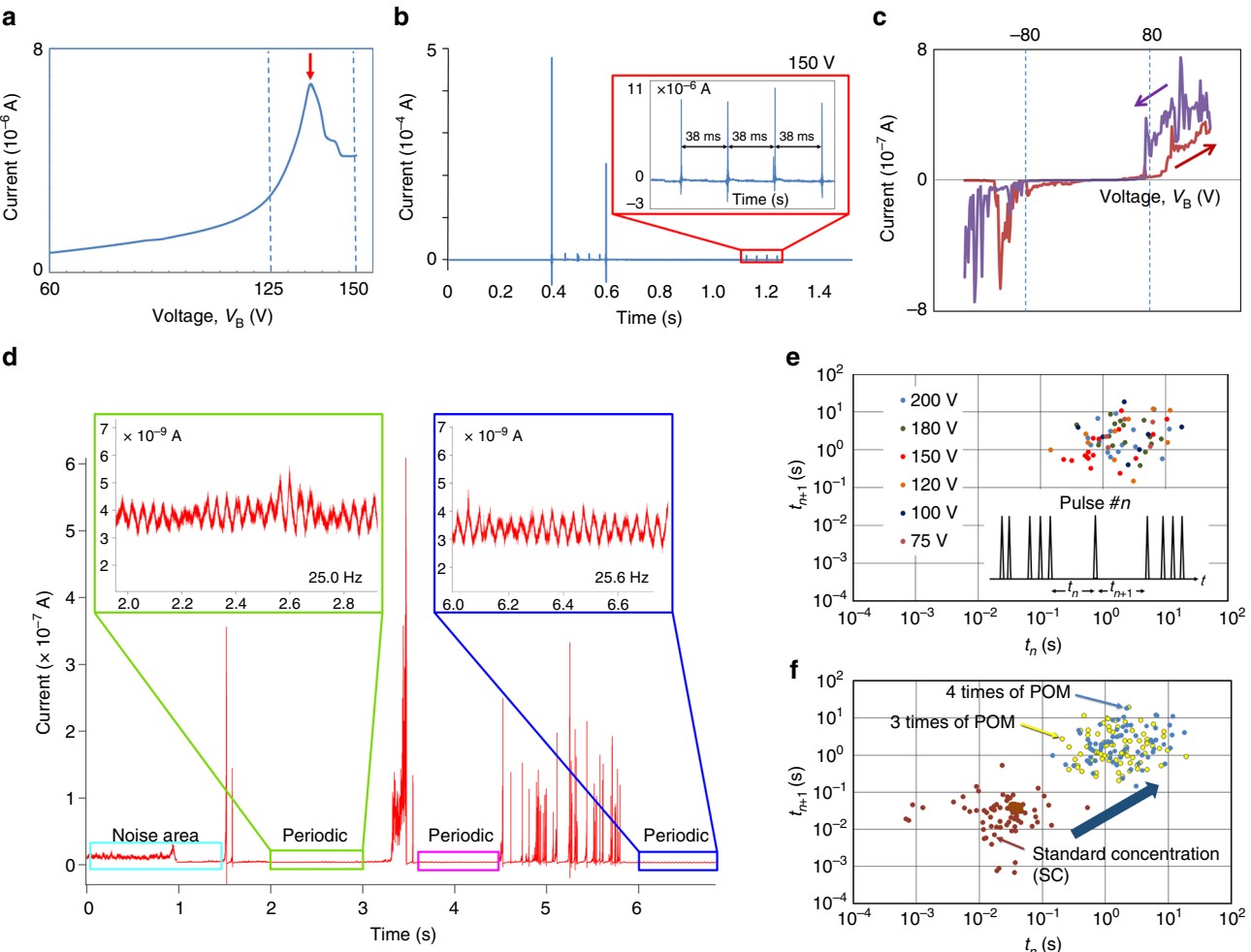

**Fig. 3** Electrical characteristics of the SWNT/POM network. **a** I–V characteristics of sample A (rinsed with ethanol) lying between electrodes (1) and (2) in Fig. 2b. **b** Plot of the measured current versus time of sample A when $V_B$ was set at 150 V. (Inset) Magnification of periodic current impulses. **c** I–V curve of the water-treated POM/SWNT samples (sample B). A number of NDR peaks are noticeable over approximately 80 V. **d** Time dependence of the current at 80 V. (Inset) Magnification of periodic base current modulation. **e** Poincaré plots formed by changing the applied voltage obtained using sample A. (inset) Example of impulse trains and sequences of the ISIs ($\cdots, t_n, t_{n+1}, \cdots$). **f** Dependence of ISIs for sample A on the concentration ratio of POM to SWNT

known that the conductance of a single molecular junction is changed by electronic and structural state fluctuations, resulting in observations of noisy current across such junctions. Conductance switching by oxidation and reduction of molecules has recently been reported[32], where the transient charging effect yields low-to-high current ratios exceeding 1000 at bias voltages of less than 1 V.

Based on the observations above, we built a POM/SWNT network model. In our model, the conductance of a junction between POMs and SWNTs changes from low to high, when the accumulated charges at a POM exceed its electron storage limit. To preserve multiple charges at a POM, the conductance of the junction must be low. Preserving a large number of charges at a POM results in a large potential difference across the junction, which induces low-to-high conductance switching.

When the electrons accumulated at a POM in the POM/SWNT network are discharged through a highly conductive junction, they are transferred to the neighbouring POM with the largest potential gradient. If the charging limit of the neighbouring POM is also exceeded, the discharge causes a chain reaction to take place in the network. A cascade of charges will occur in the network at that time and it will be detected as a charge impulse at the electrode.

**Cellular automata model of SWNT/POM network device.** A cellular automata (CA) model on a 2D regular grid, representing a random network of POM particles as cells, was constructed to understand the qualitative electrical behaviours of the SWNT/POM network device. To represent the random network on a regular grid, we introduced a defect density $D_f$. Among all of the intersection points of the regular grid, $D_f$ percent were randomly chosen and regarded as defects, while the others were regarded as POM cells. Figure 4a illustrates the CA device model represented by a 2D network of POM cells (blue spheres) with counter-electrodes (yellow bars) and SWNTs bridging the neighbouring POMs (black lines), where $D_f = 45\%$. In the network, each POM is connected to from zero (minimum, meaning it is an isolated POM) to eight (maximum) neighbouring POMs, because vacant cells (defects) are excluded. For a POM cell located at $(i,j)$ representing the discrete position in a 2D space, let $m_{i,j}$ be the number of the neighbouring POM cells. The number of charges (being proportional to the potential) of a POM particle at time $t$ is defined by $a_{i,j}(t)$. Hence, when $m_{i,j} > 0$, the potential gradients between a POM cell at $(i,j)$ and its neighbouring POMs are represented by $\Delta a_{i,j}^k$ ($k = 1, \ldots, m_{i,j}$), as shown in Fig. 4b ($m_{i,j} = 5$ in this example). The cells in the source electrode have constant charges of $V_B$,

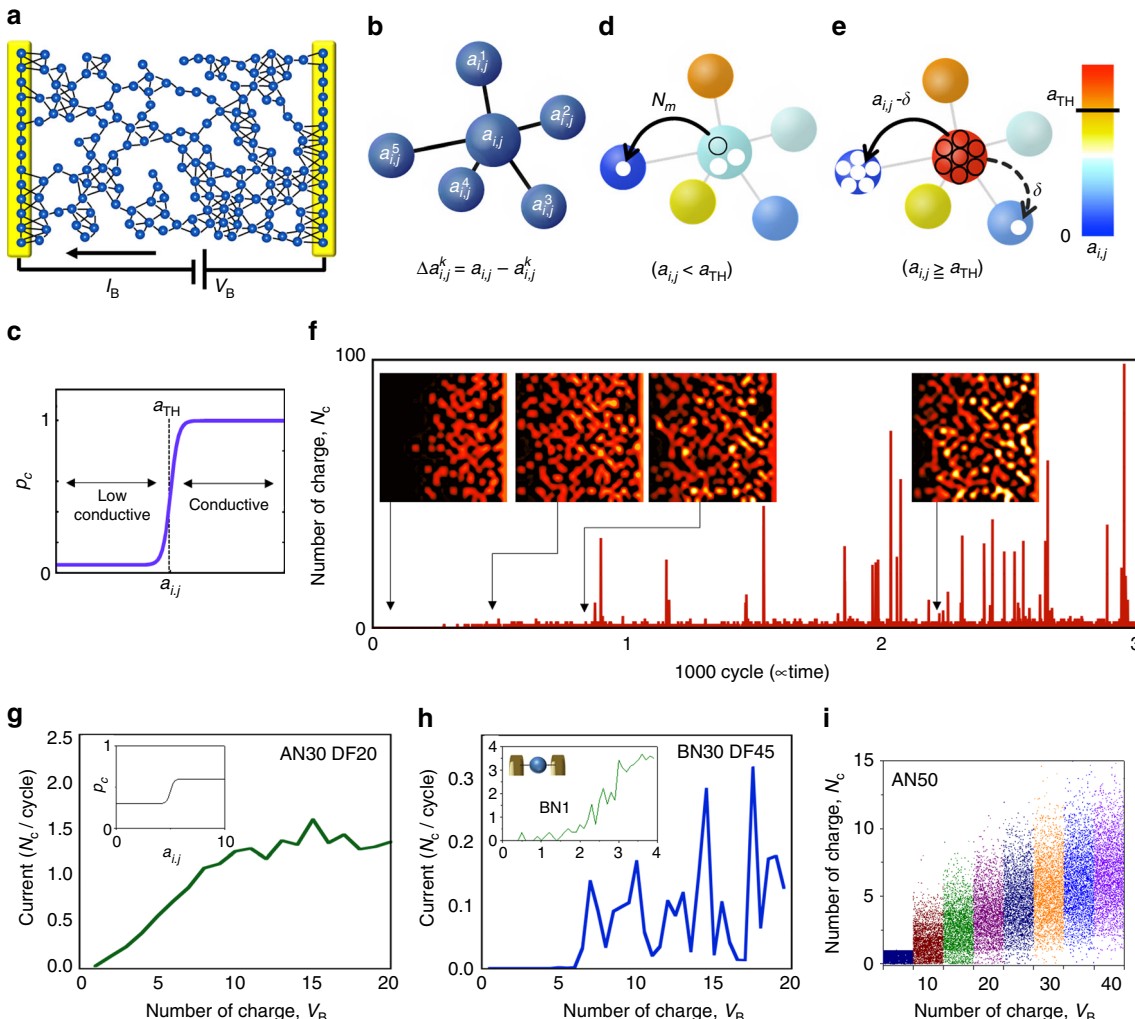

**Fig. 4** 2D CA model of the SWNT/POM network. **a** CA model on a 2D grid consisting of POM cells (blue spheres) connected with black SWNT wires and the source (S, right) and drain (D, left) electrodes. The network is on 20×14 intersection points on a 2D regular grid, and 45% of which are randomly chosen as 'detects' ($D_f = 45\%$), while the rest are the POM cells. **b** Notation of cell interaction for the state transition. The transition direction at time $t$ is selected to yield the largest gradient for state $\Delta a_{i,j}^k$ among all $k$ neighbours. **c** State transition probability showing $P_c^B$. **d, e** Schematics for two types of state transition rules applicable (**d**) when $a_{i,j} < a_{TH}$ and (**e**) for discharging when $a_{i,j} \geqq a_{TH}$. **f** Calculated impulse generation results for 3000 cycles in a 30×30 array with $P_c^A$, $V_B = 6$, $a_{TH} = 5$ and $D_f = 45\%$. Hereafter, AN30 denotes the results for a 30×30 array with $P_c^A$ and DF45 denotes $D_f = 45\%$. The inserted images are snapshots of the numbers of states in each cell at a given time. **g, h** Calculated I–V characteristics for AN30DF20 and BN30DF45, respectively, with $a_H = 5$. **g** The inset shows $P_c^A$. The definition of current is the number of charges transferred to the drain electrode $N_c$ averaged over 5000 cycles. **h** The inset shows I–V for a single cell with $P_c^B$, directly connected with the electrodes. **i** Noise generation in the base current, plotted over 5000 cycles with stepwise increment of $V_B$ for AN50 with $a_{TH} = 40$

which are supplied into the network, whereas the drain electrode is kept neutral.

In the CA model, a limited number of charges is 'stochastically' transferred from a cell to one of the neighbouring cell with the largest $\Delta a_{i,j}^k$. The probability of the charge transfer, which we call the state transition probability, is assumed by:

$$P_c\left(a_{i,j}\right) = p - \frac{q}{e^{2(a_{i,j} - a_{TH})} + 1},$$

where $a_{TH}$ represents the threshold charge, and $p$ and $q$ are the parameters of the probability, as plotted in Fig. 4c, where $p = 1$ and $q = 0.95$. Low $P_c$ suggests low conductivity charge transfer between cells and the high capacitive nature of the POM. Charge accumulation at each cell will occur representing the multi-redox nature of POM. Our particular proposition of the charge transition rule is a definition of the threshold charge $a_{TH}$ for the discharging operation. In other words, passing all of the

charges of a POM cell to a neighbouring cell means that the POM junction becomes highly conductive because of the state change of the molecule. When $a_{i,j} < a_{TH}$, a small number of charges is transferred to one of the neighbours, as shown in Fig. 4d, whereas when $a_{i,j} \geq a_{TH}$, all of the charges should be discharged and transferred to its neighbours, as shown in Fig. 4e. Due to the discharging transfer, neighbouring cells generally have an excess number of charges ($\geq a_{TH}$) at the next update cycle. The subsequent discharging results in a chain reaction (discharging-event propagation) in the 2D network, providing a charge cascade in the CA system. The current is defined as the accumulated number of charges $N_c$ transferred into the drain electrode at a given time $t$. Here, the time is divided by the calculation cycle.

We performed the 2D CA calculations under various conditions. The computational results well reproduced the experimental results, showing random spike generation, NDR properties and noise generation. Figure 4f shows the current with

respect to time for a 30×30 array with $(p, q) = (0.6, 0.3) \equiv \boldsymbol{P}_c^A$, $V_B = 30$, $a_{TH} = 5$ and $D_f = 45\%$ where $V_B$ is a dimensionless value that is proportional to the bias voltage. Random impulse generations is exhibited in the posterior half of Fig. 4f. The inset images depict the gradient of the number of charges in each cell at a given time. The right and left sides are the source and drain electrodes, respectively. During the first 100 cycles, a continuous charge gradient appears, although no current is evident. At 500 cycles, a small current is observable, as are many bright cells (having large $a_{i,j}$) on the right side. Before the first impulse, a bright cell, which exceeds $a_{TH}$, is evident at the left edge of the 2D array. After around 1000 cycles, large impulses appear with accompanying adjunctive medium and small pulses. After 2000 cycles, similar random impulse generation continues as long as $V_B$ is applied.

Figure 4g shows the current versus $V_B$, i.e. the I–V characteristics, in a 30×30 array with $\boldsymbol{P}_c^A$ and $D_f = 20\%$. The current increases monotonically as $V_B$ increases, although no large impulses are observable. Figure 4h shows the I–V characteristics when $(p, q) = (1, 0.95) \equiv \boldsymbol{P}_c^B$ (Fig. 4c). The current in Fig. 4h exhibits a lower increase at lower $V_B$ and distinct peaks at higher $V_B$. The I–V characteristics shown in Fig. 4g, h are analogous with the experimental I–V characteristics shown in Fig. 3a, c, respectively. Correspondingly, the lower (higher) $P_c$ values applied for primary charge transfer indicate the high (low) charge capability of each cell. The SWNT/POM networks under different treatments should provide different POM chemical conditions. The deionized water-treated POM must maintain higher multi-redox activity by providing the system with low conductance at lower $V_B$. The number of impulses and the threshold $V_B$ are affected by $P_c$. From the viewpoint of the calculated number of impulses against $V_B$ for different $P_c$ and array sizes, the higher threshold $V_B$ and lower number of impulses generated for $\boldsymbol{P}_c^A$ than $\boldsymbol{P}_c^B$ agrees with the threshold bias voltages trends observed for samples A and B.

The noise was also reproduced by this model. Figure 4i depicts the current at each cycle for stepwise increments of $V_B$, under the subthreshold condition of impulse generation. The amplitude of the current noise increases as $V_B$ increases. To generate large noise in the base current, large primary charge transport $\boldsymbol{P}_c^A$ and $a_{TH}$ are needed to avoid impulse generation. The use of a threshold and sufficiently high potential are required to generate impulses. If $V_B$ is lower than the threshold, impulses never appear, no matter how much time passes. With sufficient $V_B$, even in larger arrays, impulses appear after a long time. In a 300×300 array, the first impulse appears after 160,000 cycles, and the impulses have large magnitudes.

In the model, we focused on conductance switching between POMs and SWNTs induced by potential differences across their junctions. In a real POM/SWNT network, the occurrence and disappearance of highly conductive passes should be caused by complex chemical reaction phenomena. The reaction centre is electro-chemical reduction[33], i.e. multiple electron charging entailing structural change of POMs[31]. The discharging is caused by electro-chemical oxidation. Aggregation and dissociation of counter-cations, which are generally water ions, at the POM junction should play a significant role in the conductance switching between POMs and SWNTs. The chemical reaction rate should be much slower than that of charge transfer. After the total discharge, the interface becomes normal again via a reversible reaction. The POM returns to a neutral state and charge accumulation in the network starts again. The amount of charge accumulated in real POM particles, however, remains unknown. The threshold charge number, $a_{TH} = 5$, used in this study was an appropriately small for reproducing distinct impulse

generation. Smaller $a_{TH}$ increases the appearance of impulses and decreases their magnitude.

Based on the POM/SWNT device model above, we succeeded to demonstrate basic functionality of reservoir computing. Reservoir computing is a computation framework that can be found in certain dynamical recurrent neural networks[27], as well as in classical liquid state machines[34] consisting of a network of randomly connected spiking neurons. Many types of physical media have been introduced as reservoirs, such as spiking neural networks[9], photonic medium[26] and soft matter[35]. Supplementary Figure 2a depicts the typical structure of a reservoir system. The details of the simulation using the POM/SWNT model and their possible experimental setups are described in Supplementary Note 2 and 3, respectively. Although the present reservoir computer is built up on a simulated model of POM/SWNT, we are now planning to fabricate the system on real POM/SWNT devices.

## Methods

**Device fabrication**. The SWNTs were purchased from Carbon Nanotechnologies, Inc. (Houston, TX) and annealed at 200 °C for 20 h to remove the amorphous carbon. The SWNTs were washed with 12 M HCl to remove metallic catalysts and were subsequently rinsed with DI water. To prepare the SWNT/POM complex, 0.8 mg of $PMo_{12}$ was dissolved in 10 mL of ethanol, and 1.6 mg of the purified SWNTs was added to the solution. The solution was sonicated for 16 h and centrifuged at 4000 rpm for 3 h. The supernatant (1 mL) was vacuum-filtered using 1-μm-mesh nitrocellulose filter paper (MCE, Millipore) to fabricate the SWNT/POM network structure on the filter paper. The network structure was rinsed with ethanol or pure DI water. The wet filter paper was adhered to a Si substrate with a 30 or 100 nm oxidised layer with the network side facing the substrate surface. The substrate was then placed upside-down on an open-mouth bottle filled with acetone, and the bottle was heated to 80 °C. The evaporating acetone melted the filter paper, and the SWNT network was transferred to the SiO₂ surface of the substrate. Thermal deposition with a patterned metal mask was conducted to fabricate 80-nm-thick Au electrodes as shown in Fig. 2b.

**Measurements**. The electrical measurements were performed using a four-probe system with a data acquisition (DAQ) module (National Instruments, NI USB-9234), which was controlled using the LabVIEW 2014 software. The bias voltage was generated using a function generator (NF Corporation, Wavefactory WF1974) and amplified by 30 times with a voltage amplifier. The output current was transduced by a preamplifier to an output voltage, which was measured using the DAQ system. The currents at each point was integrated over 10 PLCs at 60 Hz and then averaged.

**Simulation of POM/SWNT CA model**. A $N \times N$ rectangular grid was created, and $D_f$ percent of the intersection points were randomly set as vacant (defect) cells, while the remainder were set as POM cells. As in conventional CA, the state of a POM cell at time $t$, represented by $a_{i,j}(t)$, was determined by its previous state at time $t-1$, i.e. $a_{i,j}(t-1)$, and the previous states of its neighbouring POM cells, i.e. $a_{i-1,j-1}(t-1)$, $a_{i-1,j}(t-1)$, $a_{i-1,j+1}(t-1)$, $a_{i,j-1}(t-1)$, $a_{i,j+1}(t-1)$, $a_{i+1,j-1}(t-1)$, $a_{i+1,j}(t-1)$ and $a_{i+1,j+1}(t-1)$. Our state transition rule was as follows: step 1, when $m_{i,j} > 0$, calculate the potential gradients between a POM cell located at $(i,j)$ and its neighbouring POM cells at time $t-1$, represented by $\Delta a_{i,j}^k(t-1)$, where $k$ represents the index number of the neighbours and ranges from 1 to $m_{i,j}$; step 2, find the largest potential gradient $\Delta a_{max}$ among the candidates [$= \arg\max_{k=1,\cdots,m_{ij}} \Delta a_{i,j}^k(t-1)$]; step 3, when $a_{i,j}(t-1)$ is smaller than $a_{TH}$, a limited number of charge [$N_m(\Delta a_{max})$] is stochastically transferred, obeying the state transition probability function $P_c(a_{i,j})$, to the neighbouring cell with $\Delta a_{max}$, as shown in Fig. 4d. The charge limit [$N_m(\Delta a_{max})$] has an exponential relation based on Marcus theory[36], $N_m(\Delta a_{max}) = \lceil \varepsilon e^{\gamma \Delta a_{max}} + 1 \rceil$, where $\lceil \cdot \rceil$ represents the floor function, $\varepsilon$ is the transfer coefficient and $\gamma$ is the sensitivity constant. When $a_{i,j}(t-1)$ is larger than $a_{TH}$, all of the charge moves to the two neighbours with the largest and second-largest potential gradients. The charges are moved to the neighbours on a ratio of 9:1, as shown in Fig. 4e. This charge split ratio is quite sensitive to the density and magnitude of the impulse. The array has two additional columns at each side as the source and drain electrodes. $V_B$, the number of charges in the cells of the source column, is a constant. The number of charges in the cells of the drain column is zero. The current is the accumulated number of charges transferred into the cells of the drain column at one time. In the I–V calculations, the current was accumulated every 5000 cycles. $V_B$ was derived using $\lceil \alpha V_B \rceil$; here, the coefficient $\alpha = 0.5$ was used, where $\alpha$ is necessary to control the resolution step and the floor function is used to make $V_B$ an integer.

**Data availability**. The source code for Fig. 4 and Supplementary Fig. 2, and other data that support the findings of this study are available from the corresponding author upon reasonable request.

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

## Acknowledgements

Hi.T. thanks Prof. T. Morie of Kyushu Institute of Technology for the fruitful discussions. This work was supported in part by JSPS KAKENHI Grant Numbers JP21710107, JP20111001, JP15K12109, JP20111012, JP25110002, JP16H00968 and JP25110015, and by PRESTO JST JPMJPR1521-15655977. Hi.T. also thanks the Inamori Foundation and Iketani Foundation for their financial support.

## Author contributions

Hi.T. and T.O. conceived the study and designed the experiments. Hi.T., L.H., L.F., A.T.Y., D.T. and T.O. fabricated the network devices and performed measurement system setup. Hi.T., Ha.T. and L.F. performed the ISI analysis. M.A.-K. built a cellular automaton model of the network device and numerically analysed the model with T.A. T.A. built the cellular automaton software simulator and conducted reservoir computing on the network device. All of the authors discussed the results and commented on the manuscript. Correspondence and requests for materials should be addressed to Hi.T.
