## [Peer Review File · Nature Communications]

Reviewer #1 (Remarks to the Author):

The paper reports an interesting molecular neuromorphic device that imitates the nerve impulse generation of neuronal active membranes. While the work is interesting and original, there are a few points that could use clarification.

- 1) Some information about the biological model that this device mimics would be helpful. It would also be interesting to see how the pulses generated by the device compare to the biological model.
- 2) In a few words, explain what the gap is between silicon CMOS devices and emerging non-CMOS devices.
- 3) The author claims that the device is extremely low-power. What is the power consumption? How does it compare to other devices?
- 4) A cross-sectional or 2D model of the overall device would be helpful to understand the structure.
- 5) There seem to be some inconsistencies in the figures comparing samples A & B.
 - a. Why is there no forward AND backward sweep in Fig 2d (sample A)?
 - b. Why was the sweep rate different for Fig 2f and 1b? Clarify.
 - c. It would be easier to compare the results of Fig 2g and Fig 2e if they were made to look more like one another.
 - d. Test results for sample A are shown in Fig 2i and 2f. Were these tests also done for sample B? How do the results compare?
- 6) Elaborate on the reversible chemical reaction that occurs after the total discharge.
- 7) The methods do not give a clear idea of the overall device structure (ex: how the electrodes are fabricated). Some of the testing parameters are also omitted (ex: voltage sweep rates).
- 8) Some of the figures are difficult to read.
 - a. Fig 2: Too busy. Remove non-essential information or split into multiple figures. Use different colors or shapes that are easier to distinguish in Fig 2i and 2j.
 - b. Fig 3: Too busy. Remove non-essential information or split into multiple figures.
 - c. Fig 4c: It is difficult to see the lines showing the magnified view area.

Reviewer #2 (Remarks to the Author):

The authors reported the observation of pulse generation in the POM/SWNT network, and this phenomenon was believed to be originated from the NDR with respect to the POM junction and noise generation in the whole network system. At the same time, a 2D cellular automata (CA) model representing a network of POM particles as cells was constructed to understand the qualitative electrical behaviors of the SWNT/POM network device.

As mentioned by the authors, the current-voltage (I-V) characteristics of a SWNT/POM device was measured by point-contacted current-imaging (PCI) atomic force microscope (AFM). Without POM, it is Ohmic contact between SWNT and the probe of AFM? Otherwise, the observed peaks; i.e. the current increased non-monotonically as the bias voltage increased may not be due to the NDR characteristics

of the SWNT/POM device. At the same time, the NDR positions in the I-V plots is out of control, it is likely due to the charging and discharging effect in the SWNT/POM film.

It is also opinionated for the authors to claim that POM/SWNT mixed networks is "neuromorphic network" because no neuron signal summation and processing functions were emulated or demonstrated, but only some anomalous signal noises are observed in the SWNT/POM film. Based on the above comments, it can not be accepted in Nature Communications, but it can be submitted to Scientific Reports.

Reviewer #3 (Remarks to the Author):

This paper describes experimental work on POM/SWNT networks and simulations of the networks. Both the experiments and simulations show generation of current pulses, negative differential resistance and noise characteristics and the authors conclude that both experiments and simulations produce results that are similar to the characteristics of biological neurons/synapses. This is an interesting field and if the author's assertions were proven the paper might be considered for publication in Nature Communications. However, I have 2 significant reservations about this paper:

1) Some of the fundamental assumptions made in the paper are either not justified or not explained sufficiently well:

- line 38-40: it is not clear to me that spontaneous random spike generation and NDR are characteristic neural properties. Even if we agreed that biological systems exhibit such properties, it is not clear that observation of those properties is sufficient to conclude that the system is comparable to a system of biological neurons: many other systems exhibit NDR and / or current noise in various forms but they are not claimed to be similar to neurons.

Real neurons do not generate pulses randomly, so in what way precisely is the system similar to the biological one?

- The physical basis for the model described between line 160 and 180 is not clear. The schematic in figure 3a looks like an array of nanoparticles and the model seems to be a model of electron transfer that is dependent on Coulomb blockade. But the effect of charging / the capacitance of the islands is not really explained, the role of the CNTs is not clear and the relationship of this model to the schematic in figure 2a is not clear. Similarly the physical origin of the expressions for P_c is not explained.

It appears that the model is similar to that of Elteto et al discussed in several articles in PRB/PRL more than 10 years ago (for example), but no comparison with the literature is provided. What is similar/different about this model? The authors suggest (line 219) that the generation pulses is somehow similar to the behaviour of biological neurons: does that mean that other chains of nanoparticles should be considered analogous to biological neurons? Certainly it is well-known that Coulomb blockade results in current spikes in such systems.

- The model in figure 4 appears to be essentially a model of percolation with tunnelling of electrons. How is this model similar (or different) to that described in the literature (e.g. papers by Fostner et al and Grimaldi et al in PRB in 2014, and references therein such as the hopping model of Efros and Shklovskii)?

- Many of the final statements (e.g. lines 242, 244, 247) involve the words "could" and "may". The statements would need to be much more definitive, and supported by evidence, for the article to be published.

It may be that these points can be explained, but my feeling is that the paper would need to be very significantly revised in order to do so.

2) I find that many aspects of the paper are unclear or are insufficiently well explained. In almost

every paragraph I found statements that I had difficulty understanding or where the justification was either unclear or did not seem to be correct. For example:

- line 36: what is an "electron cascading" molecular junction
- references 8, 9, 11, 13 do not appear to support the statements that the authors attributed to them e.g. on line 70 the authors mention CNT networks but the paper appears to be about BN/graphene.
- Line 75: "CNT-based conductors generate large electrical noise" is a global statement that is absolutely not true.
- Line 78: what is an "absorbent molecule"?
- Line 82: what is the justification for "possibly extremely low power" and is this consistent with line 112 where it emerges that these devices operate at voltages $> 100V$?
- line 96: what is the physical origin of the NDR peaks in figure 1b? Why are they different in each of the traces? What are the different traces in figure 1b?
- line 103: the multi terminal electrodes appear to have different lengths, but why? Why is this useful?

"Terminal distances" is not clear.

- Line 129: the current oscillations are suspiciously like some kind of "pickup". The authors do not attempt to explain them.
- Line 136: in what sense is the generated pulse "constant frequency"?
- Line 139: what exactly is Hodgkins class II behavior? What is the exact relationship between that behaviour and the experimental data?
- line 169: what is "insulative charge transfer"?
- line 190: how can it be that no current is detected at 100 cycles, and less current is detected at 500 cycles?
- Line 199: it is not at all obvious that figures 3g and 3h are analogous to figures 2d and 2f.

I emphasise that these are representative examples, and this list is not comprehensive.

Reply to Reviewer #1

- **Some information about the biological model that this device mimics would be helpful. It would also be interesting to see how the pulses generated by the device compare the to the biological model.**

The device does not mimic a pure biological neuron model, but is an integrate-and-fire neuron (IFN) model (cf. https://en.wikipedia.org/wiki/Biological_neuron_model#Integrate-and-fire), which is widely used in contemporary spiking neural networks in the field of computational neuroscience. The IFN has a nominal membrane potential, and the potential results from integrating nominal charges (postsynaptic input currents) on a nominal membrane capacitor. When the potential exceeds a given threshold, the membrane capacitor is fully discharged, meaning that all the integrated charges on the capacitor are fired. We propose that these integrate-and-fire operations are very similar to our charging (integrating) and discharging (firing) model of a single POM.

- **Why was the sweep rate different for Fig 2f and 1b? Clarify.**

(The previous Fig. 2f is now Fig. 3a. Fig. 1b retains the original numbering.)

This is now clarified in the main text, as follows. The *I-V* curves shown in Fig. 1b were obtained from one POM junction using the sweeping mode of the AFM controller, and the sweep speed was fixed. The *I-V* curves shown in Figs. 3a and 3c were obtained by a semiconductor parameter analyser with a controllable sweeping speed. The problem in the AFM measurement was due to the fact that the NDR characteristics were observed only in one polarity, as shown in Fig. 1b. Since Ref. 8 demonstrated that NDR characteristics were observed in both the polarities, we conclude that the sweep rate does not affect the results.

- **It would be easier to compare the results of Fig 2g and Fig 2e if they were made to look more like one another.**

We agree with the comment, and the impulse generation behaviour is now clearly compared in Figs. 3e and 3f (previous Figs. 2e and 2g) by the return maps.

- **Elaborate on the reversible chemical reaction that occurs after the total discharge.**

Previous studies have shown that the multi-redox of POM needs the existence of counter ions. (Reference No. 19; Kaba, M. et. al, *J. Phys. Chem.* 100, 19577-19581, (1996)) We suppose a chemical reaction of counter ions with POM occurs to cause total discharge. Because the POM has to return to the original state after discharging to maintain the original multi-redox feature, the chemical reaction has to be reversible. However, the supposed chemical reaction is still not obvious. We have included a description of the counter ions for the chemical reaction.

- **The methods do not give a clear idea of the overall device structure (ex: how the electrodes are fabricated). Some of the testing parameters are also omitted (ex: voltage sweep rates).**

The experimental methods section has been revised to address these points.

- **Some of the figures are difficult to read.**

The figures have now been improved.

Reply to Reviewer #2

- **As mentioned by the authors, the current-voltage (I-V) characteristics of a SWNT/POM device was measured by point-contacted current-imaging (PCI) atomic force microscope (AFM). Without POM, it is Ohmic contact between SWNT and the probe of AFM? Otherwise, the observed peaks; i.e. the current increased non-monotonically as the bias voltage increased may not be due to the NDR characteristics of the SWNT/POM device. At the same time, the NDR positions in the I-V plots is out of control, it is likely due to the charging and discharging effect in the SWNT/POM film.**

In a random network fabricated by SWNTs only (without POM), then Ohmic conduction would occur. NDR peaks appeared at different voltages, as shown in Fig. 1b, depending on their particle size. As mentioned in the referee's comment, we totally agree that the phenomenon is caused by the charging and discharging effects in the SWNT/POM film. Thus, we set up a simulation model of electron cascading, as shown in Fig. 4.

- **It is also opinionated for the authors to claim that POM/SWNT mixed networks is “neuromorphic network” because no neuron signal summation and processing functions were emulated or demonstrated, but only some anomalous signal noses are observed in the SWNT/POM film.**

In the revised manuscript, instead of exhibiting neuromorphic structures of POM/SWNT, we introduced a framework of neuromorphic computation, termed *reservoir computing*, which can be found in a type of dynamical recurrent neural networks. We performed neuromorphic reservoir computing on the POM/SWNT device model, and demonstrated *temporal coding* of nonlinear sequences on the device model with external controllers implementing the FORCE learning algorithm (Fig. 5). This clearly shows the applicability of our use of the term neuromorphic network.

Reply to Reviewer #3

- **Some of the fundamental assumptions made in the paper are either not justified or not explained sufficiently well:**

- **line 38-40: it is not clear to me that spontaneous random spike generation and NDR are characteristic neural properties. Even if we agreed that biological systems exhibit such properties, it is not clear that observation of those properties is sufficient to conclude that the system is comparable to a system of biological neurons: many other systems exhibit NDR and / or current noise in various forms but they are not claimed to be similar to neurons.**

Real neurons do not generate pulses randomly, so in what way precisely is the system similar to the biological one?

(The previous line 36 is shifted to the present page 4 lines 12-14.) The *I-V* curves, including NDR characteristics shown in Fig. 1a, prove that electrons can be held and released (charged and discharged) at a POM site by changing the applied voltage. Based on the phenomena, we performed numerical simulations, as shown in Fig. 4, and proved that the impulse generation resulted from the electrical properties of the POM. Thus, we revised the manuscript at that point, to avoid misleading readers.

- **The physical basis for the model described between line 160 and 180 is not clear. The schematic in figure 3a looks like an array of nanoparticles and the model seems to be a model of electron transfer that is dependent on Coulomb blockade. But the effect of charging / the capacitance of the islands is not really explained, the role of the CNTs is not clear and the relationship of this model to the schematic in figure 2a is not clear. Similarly, the physical origin of the expressions for P_c is not explained.**

- **It appears that the model is similar to that of Elteto et al discussed in several articles in PRB/PRL more than 10 years ago (for example), but no comparison with the literature is provided. What is similar/different about this model?**

Actually, our POM/SWNT network model was inspired by the 2-dimensional Coulomb blockade (CB) network models, which have been studied by Elteto, *et al.* and by Middleton and Wingreen, (Phys. Rev. Lett. 71, 3198 (1993)). Although both systems are quite similar in appearance with respect to their network schemes, our network system is fundamentally different from the CB system. The CB system consists of conductive islands with small capacitance connected with tunnelling paths, whereas our SWNT/POM network consists of highly capacitive POM islands connected with conductive SWNTs. The small capacitance of conductive islands in CB array plays a role in charge blockade, whereas the large capacitance of the POM plays a role to accumulate charges rather than blockade them. The P_c defined in this article is essentially similar to redox activity of the molecule rather than the capacitance. A significant difference is the existence of the definition of total discharge for the POM cell, which provides the impulse generation. Moreover, calculation of the CB system is driven by minimization of total Coulomb energy, whereas our automata model is driven by local charge number differential and stochastic provision for the direction and number of charge transfer.

- **The model in figure 4 appears to be essentially a model of percolation with tunneling of electrons. How is this model similar (or different) to that described in the literature (e.g. papers by Fostner et al and Grimaldi et al in PRB in 2014, and references therein such as the hopping model of Efros and Shklovskii)?**

Although the model in Fig. 4 (in the previous manuscript) has been removed, the significant difference is the existence of total discharging from the cell as mentioned above. Under the condition without discharging, our calculation results for SWNT/POM network provide noisy current without impulse generation as shown in Fig. 4i. The charge transport in both CB systems and hopping models and our SWNT/POM network (without total discharging) must have some physically essential analogies, because they can be described by a similar charge percolation system.

- **The authors suggest (line 219) that the generation pulses are somehow similar to the behaviour of biological neurons: does that mean that other chains of nanoparticles should be considered analogous to biological neurons? Certainly, it is well-known that Coulomb blockade results in current spikes in such systems.**

(The previous line 219 is shifted to page 10 line 19.)

It is considered that nerve impulses (spikes) of biological neurons result from nonlinear integration of neurochemical ions in the cell membrane. We define the integration of the charges and their total ejection in and from the POM for the origin of impulse generation. Furthermore, the SWNT/POM network sometimes showed periodic impulse generation, as shown in Figs. 3e and 3f, although our simulation does not reproduce such periodicity. The periodic impulse generation is a peculiar feature of biological neurons. We believe that the impulse generation system on our SWNT/POM network and other noise generation systems with shot-noise generation (such as the CB system) must have essential differences.

- **Many of the final statements (e.g. lines 242, 244, 247) involve the words “could” and “may”. The statements would need to be much more definitive, and supported by evidence, for the article to be published.**

These statements have now been revised and supported by evidence. To improve the arguments, we tried to avoid using the words “could” or “may”.

- **I find that many aspects of the paper are unclear or are insufficiently well explained. In almost every paragraph I found statements that I had difficulty understanding or where the justification was either unclear or did not seem to be correct. For example:**

- **line 36: what is an “electron cascading” molecular junction**

(The previous line 36 is shifted to the present page 2 line 12.)

We agree that “electron cascading molecular junction” was an inadequate term to explain our system. We revised a sentence in the abstract to “We propose an electron-cascading model in the SWNT/POM network consisting of heterogeneous molecular junctions that shows good agreement with the experimental results.” In addition, the entire paper has been edited and proofread by a professional scientific English editing service.

- **References 8, 9, 11, 13 do not appear to support the statements that the authors attributed to them e.g. on line 70 the authors mention CNT networks but the paper appears to be about BN/graphene.**

We would like to apologise for the errors in the list of references. These have now been corrected.

- **line 75: “CNT-based conductors generate large electrical noise” is a global statement that is absolutely not true.**

We intended to refer to the noise enhancement of the SWNT network. To avoid misleading the readers, the main text has now been modified.

- **line 78: what is an “absorbent molecule”?**

The statement “one of the reported absorbent molecule” was revised to “the adsorbed molecule.”

- **line 96: what is the physical origin of the NDR peaks in figure 1b? Why are they different in each of the traces? What are the different traces in figure 1b?**

(The previous line 96 is shifted to page 5 line 1.)

NDR occurs because of redox of POM during current pass through the nanoparticle. The peaks in Fig. 1b appear at different voltage because of the particle size, which varies different capacitance and resistance when current is passing through the nanoparticles. These are reproducible results, and always occurred in the SWNT/POM random network devices at a certain humidity.

- **“Terminal distance” is not clear.**

This has been changed to “gap distance between electrodes”.

- **line 129: the current oscillations are suspiciously like some kind of “pickup”. The authors do not attempt to explain them.**

(The previous line 129 is shifted to page 6 line 19.)

These were reproducible results, and always occurred in the wet SWNT/POM random network devices.

- **line 169: what is “insulative charge transfer”?**

(The previous line 169 is shifted to page 8 line 17.)

The “**insulative**” has been changed to “**low conductivity**” in the manuscript and Fig. 4. We defined P_c as the probability of charge transfer of a cell to the surrounding cells, and therefore, a low P_c results in lower current between the electrodes.

- **line 190: how can it be that no current is detected at 100 cycles, and less current is detected at 500 cycles?**

(The previous line 190 is shifted to page 9 line 13.)

We revised the word “less” to “small”: “At 500 cycles, a small current was detected.”

- **line 199: it is not at all obvious that figures 3g and 3h are analogous to figures 2d and 2f.**

(The previous line 199 is shifted to page 9 line 22.)

A common feature of the I - V characteristics in Figs. 3a and 4g (previously Figs. 2d and 3g) is a monotonic increase of current with voltage increase, whereas a common feature of the I - V curves in Figs. 3c and 4h (previously Figs. 2f and 3h) is a quite low current in the lower voltage range and large spikes in the higher voltage range. We thought that these are obvious differences showing analogies between Figs. 3a and 4g, and Figs. 3c and 4h. The smoothness of Figs. 3a and 4g appears to be different. This is due to the coarse increments of V_B in the I - V simulation conditions; *i.e.*, the V_B increased in steps of 5 from 0 to 10, because the V_B must be an integer number and single step increments were not sufficient to provide enough resolution of the I - V data. Therefore, the stairs appear in the lower V_B range in Fig. 4g. In the previous manuscript, the V_B was not corrected by the correction coefficient of $\alpha = 0.2$. We would like to thank you for pointing this out.

- **I emphasize that these are representative examples, and this list is not comprehensive.**

We would like to thank you for your helpful and valuable comments. We have extensively revised our manuscript after serious reflection on your comments, and those of the other reviewers. We hope you find that it is now much improved.

Reviewers' comments:

Reviewer #1 (Remarks to the Author):

This paper discusses a SWNT/POM network that generates spontaneous current impulses, mimicking nerve impulse generation. This topic would be of interest to others in the community and the wider field, and will influence thinking in the field. The authors have made appropriate edits to the manuscript to make the work convincing. Based on the information provided in the Methods section, it seems as though one should be able to reproduce the work.

Reviewer #2 (Remarks to the Author):

In this manuscript, authors reported two terminal devices with a complex film of single-walled carbon nanotubes (SWNT) complexed with polyoxometalate (POM). Some NDR peaks and irregular current peaks were observed at high voltage of 80V and 150V, which is likely due to the stochastic hopping of electrons and electron-trap mechanisms in the network of SWNT/POM. Irregular NDR peaks usually can be observed in complex film at high voltage, but no time-dependent dynamic behavior is measured and investigated.

In this manuscript, traditional neuromorphic device behaviors such as STDP, or PPF were not demonstrated. The reported device size (several millimeters) is too big, which is impossible for high density intergation, and the operation voltage of the device is too high, which is not good for low-power neuromorphic computing.

At last, the authors claimed that reservoir computing (temporal coding of nonlinear sequences) based on POM/SWNT network model were demonstrated, it seems that simulation were performed, and no experimental results were obtained. As reported by the authors, only two terminal device were fabricated in this manuscript, so what or how reservoir computing was realized?? What is the input signal and what is the output signal?

In conclusion, this manuscript can not be accepted for publication, and it is much better for Scientific Reports publication.

Reviewer #4 (Remarks to the Author):

* The proposed realization of a spiking neural model on the SWNT/POM network is a welcome contribution: many other hardware realizations of neural functionality in the literature use an analog non-spiking representation of neural signals, though often a rudimentary spiking functionality is included in those realizations. However, the wording by the authors on lines 50-51 that the list of hardware neural processors in lines 45-48 are not based on neuroscience appears to be too strong, especially since the list also includes two references ([5] and [6]) of analog/digital neuromorphic circuits on line 47. I would rather state that most of the hardware in lines 45-48 have an underlying von Neumann architecture, which makes them less efficient in terms of speed and energy, as compared to hardware implementations of devices and circuits that directly mimic neural functionality.

Also, in lines 53-55 it is stated that artificial spiking neurons that mimic nerve impulse generation [...] are essential. Though I basically agree with this statement, it needs a better motivation to convince the reader. I recommend that the authors shortly describe in a few lines what neural functionality

would be impossible when neurons would not have spiking behavior. In other words, why is spiking behavior really necessary in a neural system? What functionalities does it enable?

* I recommend to use the phrase “rudimentary computing ability” instead of “excellent computing ability” in the abstract when the authors refer to their proposed SWNT/POM network, because that describes the work better. This is not necessarily a negative point, because the proposed work has potential for high integration densities, but the presented results do not yet justify to speak of “excellent computing ability.”

* The authors claim that their molecular neuromorphic network is extremely dense. This is probably true, but please, give a concrete number how dense it actually is.

* In lines 63-70 two types of devices are identified as necessary for neuromorphic functionality. While a neuronal membrane (neuron) device is proposed in the paper, a synaptic device is referred to in reference [9], suggesting that both types of devices can be combined in a neuromorphic architecture. It would be helpful if the authors would include a discussion how synaptic functionality can be combined with their proposed spiking networks. The synapses proposed in reference [9], while based on Carbon Nanotubes, tend to have lower densities than the neural membranes proposed by the authors, so they may not be the optimal choice. Please, discuss these and related issues.

* In lines 85-86 the authors speak of “excellent learning ability” of their network. As said above, the paper does not appear to support the claim that the physical realization of the network is capable of learning. The claim of learning ability derives mostly from the simulation of reservoir computing, in which learning takes place at connections between the reservoir and the output. Furthermore, I would rather call this learning ability “rudimentary”, because the network is basically trained to follow a 1-dimensional value (in other words, it is trained to act like a wire with a delay).

* In lines 135 and 139 the authors speak of a “chaotic analysis method.” Please, provide more details on this method.

* A problem with the paper is that no concrete physical model is described that accounts for the spiking behavior of the SWNT/POM network. The equation following line 174 describes a transition probability of charges jumping between POM particles, but it is unclear where this equation originates from. Please, give a better motivation for this equation, either through a reference to literature, or through deriving it. The fact that transfer of charges is claimed to be probabilistic indicates quantum tunneling, but on the other hand the authors speak of conduction of charges. Did the authors consider the Sandpile model to describe the phenomenon that charges are transferred once they exceed a certain threshold (lines 158-160)? This model is also considered relevant for neural systems (see for example [J.M. Beggs, D. Plenz, “Neuronal Avalanches in Neocortical Circuits”, *J. Neuroscience* 23(35), pp. 11167-11177 (2003)], [J. Ouellette, “Sand Pile Model of the Mind Grows in Popularity”, *Quanta Magazine*, 7 April 2014]). One suggestion to confirm whether the Sandpile model applies is to check whether the spiking data follows the power law.

* Simulation of the 2D cellular automaton: The cellular automaton has cells arranged in a regular grid while the SWNT/POM network cells are randomly distributed. How does this difference affect the simulation results? The cellular automaton simulation results reproduce the experimental results well, as claimed in line 189, but wouldn't it be better to just simulate a random grid that more closely resembles the physical realization?

* Minor point in lines 219-220: Consider to rewrite the sentence to “The use of a threshold and a sufficiently high potential are necessary to generate spikes.”

- * Lines 224-228: The authors describe a chemical reaction being responsible for discharges of POMs in a POM-SWNT network, but what is the evidence for this assertion, and could they give more details about the chemical reaction?
- * Simulation of Reservoir Computing is based on the cellular automaton, but in reservoir computing the connectivity inside the reservoir is typically random, while in the cellular automaton it is regular. How does this affect your results? Again, it would probably be better to have a more randomly connected model closer to the physical implementation. Related to this, why isn't there a Methods description of the reservoir computing simulations? Is the description in the text sufficient?
- * Minor point: Reservoir computing refers to a more general model, which includes the Liquid State Machine. The Liquid State Machine was originally considered with spiking neural dynamics in mind.
- * The list of references is well-chosen, and is helpful for the reader.
- * Fig. 1a: There is no reference in the text to this figure. It is a bit difficult to estimate the diameter of the SWNTs. Is it correct to conclude that they are approximately 20nm?
- * Fig. 1b: What is the meaning of the different curves. Please, use inset or describe in the caption.
- * In conclusion: While the paper's proposal has a good potential for high-density realizations of spiking neuromorphic hardware, it needs more work. In particular I would like to encourage the authors to look into the Sandpile model as the underlying mechanism of the observed spiking behavior. The simulations should be improved to account for the random layout of the POM cells, but it probably does not take too much efforts to adjust the source code to accomplish this. It would be good if the authors would be more specific on how learning functionality could be realized in the physical implementation (a discussion would suffice).

Reply to Reviewer #2:

- **In this manuscript, traditional neuromorphic device behaviours such as STDP, or PPF were not demonstrated. The reported device size (several millimetre) is too big, which is impossible for high density integration, and the operation voltage of the device is too high, which is not good for low-power neuromorphic computing.**

The term ‘neuromorphic’ includes wide-level of abstraction of neuronal systems, from the neurophysiological to the behavioural level. For example, STDP synapses, spiking neurons, and networks of those neurons and synapses, are primary implementation targets in modern neuromorphic engineering. In this manuscript, we focused on spiking neurons and their complex networks only as morphing targets, because the described POM/SWNT device is the first molecular electronic device in the world to exhibit spiking behavior. Of course, targeting synaptic/learning behaviours, including STDP, is important, as the reviewer suggested. Since it requires the introduction of completely different types of molecules, *i.e.*, memristive/synaptic molecules, into the present POM/SWNT network, we believe this investigation should be continued in a separated work.

Recently we found that a micro-scale POM/SWNT complex device with a 1 μm gap also generated impulses at low bias voltages (0.2–1 V). The device was fabricated by employing controllable electrophoresis to align the SWNT. We added the preliminary experimental results of the POM-SWNT device in the Supplementary Information (Section S1).

- **At last, the authors claimed that reservoir computing (temporal coding of nonlinear sequences) based on POM/SWNT network model were demonstrated, it seems that simulation were performed, and no experimental results were obtained. As reported by the authors, only two terminal devices were fabricated in this manuscript, so what or how reservoir computing was realized?? What is the input signal and what is the output signal? In conclusion, this manuscript cannot be accepted for publication, and it is much better for Scientific Reports publication.**

We added descriptions of experimental setups that could be employed to realize a reservoir system involving the POM/SWNT complex, although they remain unverified, in the Supplementary Information (Section S3).

Reply to Reviewer #4:

- **The proposed realization of a spiking neural model on the SWNT/POM network is a welcome contribution: many other hardware realizations of neural functionality in the literature use an analog non-spiking representation of neural signals, though often a rudimentary spiking functionality is included in those realizations. However, the wording by the authors on lines 50-51 that the list of hardware neural processors in lines 45-48 are not based on neuroscience appears to be too strong, especially since the list also includes two references ([5] and [6]) of analog/digital neuromorphic circuits on line 47. I would rather state that most of the hardware in lines 45-48 have an underlying von Neumann architecture, which makes them less efficient in terms of speed and energy, as compared to hardware implementations of devices and circuits that directly mimic neural functionality.**

The authors completely agree with the comments above. Thus, we reorganized the section by introducing present, mainstream Neumann-based AI accelerators first and then explaining the

main points and importance of the non-Neumann-based neuromorphic chips in comparison to the accelerators (page 2, lines 45–56).

- **Also, in lines 53-55 it is stated that artificial spiking neurons that mimic nerve impulse generation [...] are essential. Though I basically agree with this statement, it needs a better motivation to convince the reader. I recommend that the authors shortly describe in a few lines what neural functionality would be impossible when neurons would not have spiking behavior. In other words, why is spiking behavior really necessary in a neural system? What functionalities does it enable?**

Coding neuronal information using spike is functionally important when transmitting actions on neuronal membranes (active transmission lines) in noisy and unreliable environments^[R1]. The computational functionality of spiking neural networks in practical applications remains unclear; however, it has recently been shown that complex and spontaneous dynamics generated by large-scale spiking neural networks are useful for blind source separation^[R2], reservoir computing, and so on. These descriptions has been inserted on page 3, lines 57–62.

^[R1]Ochab-Marcinek, A., Schmid, G., Goychuk, I. & Hänggi P. Noise-assisted spike propagation in myelinated neurons. *Phys. Rev. E* 79, 011904-(1-7) (2009).

^[R2]Hiratani, N. & Fukai T. Mixed signal learning by spike correlation propagation in feedback inhibitory circuits. *PLOS Computational Biology*, 11, e1004227-(1-36), (2015).

- **I recommend to use the phrase “rudimentary computing ability” instead of “excellent computing ability” in the abstract when the authors refer to their proposed SWNT/POM network, because that describes the work better. This is not necessarily a negative point, because the proposed work has potential for high integration densities, but the presented results do not yet justify to speak of “excellent computing ability.”**

Thank you very much for your suggestion. The authors fully agree with it, and the abstract and main text have been revised accordingly (page 2, line 36; page 4, line 92).

- **The authors claim that their molecular neuromorphic network is extremely dense. This is probably true, but please, give a concrete number how dense it actually is.**

Emerging nano-scale technologies are pushing fabrication limits. For example, a 30-nm pitch crossbar array with 900 G/cm² density has been demonstrated on a 2D surface^[R3]. It may be that the ultimate integration of networks consisting of nanometre-scale materials must be one or two orders of magnitude higher, 10–100 T/cm², than that of the high-end, fine-processing-technology product. However, the importance of the use of nanomaterials will lie not only in their dense aggregation, but also in the utilization of their natural/intrinsic properties, such as randomness, self-aggregation process, and so on, in the future novel devices.

^[R3] Khat, A., Ayliffe, P. & Prodromakis, T. High density crossbar arrays with sub-15 nm single cells via liftoff process only. *Scientific Reports*, 6, 32614 (2016).

- **In lines 63-70 two types of devices are identified as necessary for neuromorphic functionality. While a neuronal membrane (neuron) device is proposed in the paper, a synaptic device is referred to in reference [9], suggesting that both types of devices can be combined in a neuromorphic architecture. It would be helpful if the authors would include a discussion how synaptic functionality can be combined with their proposed spiking**

networks. The synapses proposed in reference [9], while based on Carbon Nanotubes, tend to have lower densities than the neural membranes proposed by the authors, so they may not be the optimal choice. Please, discuss these and related issues.

Our device imitates the spiking behaviors of complex networks; however, it does not include any synaptic function. One of the difficult challenges is to incorporate synaptic functions into the POM/SWNT networks for effective neuromorphic demonstration. One possible means of achieving this objective is to complex memristive molecules, such as BPDN molecules^[R4], in the POM/SWNT network so that POMs (as spiking neurons), memristive molecules (as synapses), or both can be localized at the SWNT junctions; however, it might take a very long time to find computational and useful functions. On the other hand, in our reservoir framework, the reservoir itself (POM/SWNT networks as complex spiking networks) and external synaptic readout wires are separated. Consequently, one can obtain one of the useful reservoir functions; *i.e.* temporal coding of complex time series, by controlling the synaptic weight through FORCE learning, while fully utilizing the complex dynamics of the POM/SWNT network. We inserted the considerations above into Supplementary Information (Section S2).

^[R4]Blum, A.S., Kushmerick, J.G., Long, D.P., Patterson, C.H., Yang, J.C., Henderson, J.C., Yao, Y., Tour, J.M., Shashidhar, R. & Ratna, B.R. Molecularly inherent voltage-controlled conductance switching. *Nature Materials*, 4, 167-172, (2005).

• In lines 85-86 the authors speak of “excellent learning ability” of their network. As said above, the paper does not appear to support the claim that the physical realization of the network is capable of learning. The claim of learning ability derives mostly from the simulation of reservoir computing, in which learning takes place at connections between the reservoir and the output. Furthermore, I would rather call this learning ability “rudimentary”, because the network is basically trained to follow a 1-dimensional value (in other words, it is trained to act like a wire with a delay).

Thank you very much for your suggestion. The authors fully agree with it, and the abstract and the main text have been revised accordingly (page 2, line 36; page 4, line 92).

• In lines 135 and 139 the authors speak of a “chaotic analysis method.” Please, provide more details on this method.

The details, not only of the method itself^[R5], but also of its meanings, have been added (page 7, lines 144–154).

^[R5]Huikuri, H.V., Mäkikallio, T.H., Peng, C.K., Goldberger, A.L., Hintze, U. & Møller, M. Fractal correlation properties of R-R interval dynamics and mortality in patients with depressed left ventricular function after an acute myocardial infarction. *Circulation*, 101, 47-53, (2000).

• A problem with the paper is that no concrete physical model is described that accounts for the spiking behavior of the SWNT/POM network. The equation following line 174 describes a transition probability of charges jumping between POM particles, but it is unclear where this equation originates from. Please, give a better motivation for this equation, either through a reference to literature, or through deriving it. The fact that transfer of charges is claimed to be probabilistic indicates quantum tunneling, but on the other hand the authors speak of conduction of charges. Did the authors consider the sand pile model to describe the phenomenon that charges are transferred once they exceed a

certain threshold (lines 158-160)? This model is also considered relevant for neural systems (see for example [J.M. Beggs, D. Plenz, “Neuronal Avalanches in Neocortical Circuits”, *J. Neuroscience* 23(35), pp. 11167-11177 (2003)], [J. Ouellette, “Sand Pile Model of the Mind Grows in Popularity”, *Quanta Magazine*, 7 April 2014]). One suggestion to confirm whether the sand pile model applies is to check whether the spiking data follows the power law.

Regarding the question ‘Did the authors consider the Sand pile model to describe the phenomenon that charges are transferred once they exceed a certain threshold (lines 158–160)?’, the answer is no. Indeed, we were not aware of the sand pile model upon constructing the POM/SWNT model. We agree that, as you mentioned, the description of the spiking behavior of the SWNT/POM network based on our model was insufficient from a physical perspective. The model was inspired by the multi-redox properties of POMs as well as the drastic conductance change of a single molecule through charging. Our transition probability equation (page 9, between lines 204–205) is intended to define the conductance change at a POM junction with a charging threshold. The descriptions above have been added in the revised manuscript (page 8, lines 162–179).

The relevance of the sand pile model to spiking neural networks was incredibly interesting to us, because our model is based only on pass-out and threshold systems. We numerically checked the appearance of the power law in our network model, by following the method described in a paper the reviewer suggested, [J. M. Beggs, D. Plenz, “Neuronal Avalanches in Neocortical Circuits”, *J. Neuroscience* 23(35), pp. 11167–11177 (2003)]. Consequently, we observed a similar power law, as shown in Fig. R2. They are the numerical simulation results of a random network, shown in the revised Fig. 4a, with $N = 30$, $a_{TH} = 5$, $V_B = 6$, and $D_f = 45\%$. Figure R1 shows spike raster plots, where each plot represents a cell that has more than the threshold charges (electrons) and is then discharged. The exponents (-2.0 for the lifetime and -1.5 for the avalanche size) were varied by changing the simulation conditions. No clear avalanche-like chain reaction is observable on the network, but all of the spikes gradually and randomly propagate toward the drain side. We further revealed that the firing history (temporal area in which the spikes propagated) in the 2D space is very important in making the space conductive temporally, which might contribute to the exploration of short term plasticity on this device.

Figure R1 | Raster plot of the POM/SWNT network.

Figure R2 | Histogram of lifetime and size.

- **Simulation of the 2D cellular automaton:** The cellular automaton has cells arranged in a regular grid while the SWNT/POM network cells are randomly distributed. How does this difference affect the simulation results? The cellular automaton simulation results reproduce the experimental results well, as claimed in line 189, but wouldn't it be better to just simulate a random grid that more closely resembles the physical realization?

We performed numerical simulations using the improved 2D CA model, as shown in the revised Figs. 4 and S2. Following the model change, the descriptions in the main text (page 9, lines 186–201), and the figures and their captions (Figs 4a–e, 4f-h, S2b–f). As can be seen in the revised Figs. 4 and S2, the results without defects (in our previous manuscript) and without defects exhibit no significant differences.

- **Minor point in lines 219-220:** Consider to rewrite the sentence to “The use of a threshold and a sufficiently high potential are necessary to generate spikes.”

Thank you very much. We changed the original sentence to the recommended one. (page 11, lines 250–251)

- **Lines 224-228:** The authors describe a chemical reaction being responsible for discharges of POMs in a POM-SWNT network, but what is the evidence for this assertion, and could they give more details about the chemical reaction?

In the model, we focused on conductance switching between POMs and SWNTs induced by potential difference across their junctions. In a real POM/SWNT network, the occurrence and disappearance of highly conductive passes should be caused by complex chemical reaction phenomena. The reaction centre is electro-chemical reduction^[R6], *i.e.* charging multiple electrons entailing structural changes of POMs^[R7]. The discharging is caused by electro-chemical oxidation. Aggregation and dissociation of counter-cations, which are generally water ions, at the POM junction should play a significant role of the conductance switching between POMs and SWNTs. The descriptions above have been added in the revised manuscript (page 12, lines 255–261).

^[R6] T. Okujima, H. Matsumoto, S. Mori, T. Nakae, M. Takase, H. Uno, Synthesis of cyclo[8]pyrrole-polyoxometalate double-decker complex, *Tetrahedron Lett.* 57, 3160-3162 (2016).

^[R7]H. Wang; S.Hamanaka; Y. Nishimoto; S. Irlle; T. Yokoyama; H. Yoshikawa; K. Awaga In Operando X-ray Absorption Fine Structure Studies of Polyoxometalate Molecular Cluster Batteries: Polyoxometalates as Electron Sponges, *Journal of the American Chemical Society.* 134, 4918–4924 (2012)

- **Simulation of Reservoir Computing is based on the cellular automaton, but in reservoir computing the connectivity inside the reservoir is typically random, while in the cellular automaton it is regular. How does this affect your results? Again, it would probably be better to have a more randomly connected model closer to the physical implementation. Related to this, why isn't there a Methods description of the reservoir computing simulations? Is the description in the text sufficient?**

Thank you very much for the useful comments. The authors introduced random defects inside the reservoir, and re-simulated it. No significant changes in the reservoir performances were observed. Further, due to the increase of the reservoir descriptions (although the main topics of this manuscript are exhibiting emergence of spiking behaviours in the POM/SWNT device, and considering the origin of the spike generation through device-aware cellular-automata modeling), we moved the reservoir contents to Supplementary Information (Section S2), where the aims, results, and methods are separated.

- **Minor point: Reservoir computing refers to a more general model, which includes the Liquid State Machine. The Liquid State Machine was originally considered with spiking neural dynamics in mind.**

Yes, this statement is true. Upon introducing reservoir computing, the authors added descriptions of liquid state machines^[R8] as an early model of reservoir computing, on page 12, lines 269–271.

^[R8]Maass, W., Natschläger, T. & Markram H. Real-time computing without stable states: A new framework for neural computation based on perturbations. *Neural Computation*, 14, 2531-2560, (2002).

- **Fig. 1a: There is no reference in the text to this figure. It is a bit difficult to estimate the diameter of the SWNTs. Is it correct to conclude that they are approximately 20nm?**

The estimated diameters of the SWNT and POM complex have been included in the main text (page 5, lines 97–100).

- **Fig. 1b: What is the meaning of the different curves. Please, use inset or describe in the caption.**

As stated in the main text (page 5, lines 100–111), I - V curves were obtained for different POM particles with different sizes on the SWNT. As peaks in I - V curves indicate redox potential, the curves show that differently sized POM particles have different redox potentials, as was also reported following our previous research^[R9]. Although numerous curves with differently sized POM particles were obtained in this experiment, only a few presentative curves are presented in this article.

^[R9]Hong, L., Tanaka, H. & Ogawa, T. Rectification direction inversion in a phosphododecamolybdic acid/single-walled carbon nanotube junction. *J. Mater. Chem. C* 1, 1137–1143, (2013).

REVIEWERS' COMMENTS:

Reviewer #4 (Remarks to the Author):

The authors have revised their manuscript extensively, taking into account the reviewers' comments. The manuscript is much more mature now. In principle I think the manuscript is ripe for acceptance. Below I give a few comments, and I suggest that the authors take them into account to improve readability at some places in the text.

L 36-38: I suggest rewriting the sentence to: "Rudimentary learning ability of the SWNT/POM network is illustrated by introducing a reservoir computing framework, which utilizes spiking dynamics and a certain degree of network complexity."

L 38: "indicate a possibility" -> "indicate the possibility"

L 58-60: I suggest to rewrite the sentence to: "The usefulness of spiking neural networks in practical applications has not become completely clear; ... "

L 65: "of the human brains" -> "of human brains"

L 91-93: I suggest rewriting to: "In the Supplementary Information (Section S2) we illustrate the potential of the SWNT/POM network model for neuromorphic reservoir computing by demonstrating basic learning ability of the network."

L 110: "is not yet" -> "has not yet"

L 221: "well" occurs twice in sentence. Delete first occurrence.

General: the authors have investigated the sandpile model as I suggested in my previous review, and though they obtained some interesting results from simulations, it appears that there is not sufficient evidence to explain the spiking behavior of the proposed network in terms of this model, so they have left this element out of the paper. This is basically fine with me. Hopefully, the sandpile model will act as an inspiration for their future work.

Reference [R3] of Khiat et al in the Replies to Reviewers does not appear in the paper itself. Please, check whether this is indeed what you intend.

We revised the article by following reviewer#4's comments as following.

Reviewer's Comment:

L 36-38: I suggest rewriting the sentence to: "Rudimentary learning ability of the SWNT/POM network is illustrated by introducing a reservoir computing framework, which utilizes spiking dynamics and a certain degree of network complexity."

Answer:

L 38: "indicate a possibility" -> "indicate the possibility"

Reviewer's Comment:

L 58-60: I suggest rewriting the sentence to:

"The usefulness of spiking neural networks in practical applications has not become completely clear; ... "

Answer:

L 65: "of the human brains" -> "of human brains"

Reviewer's Comment:

L 91-93: I suggest rewriting to:

"In the Supplementary Information (Section S2) we illustrate the potential of the SWNT/POM network model for neuromorphic reservoir computing by demonstrating basic learning ability of the network."

Answer:

L 110: "is not yet" -> "has not yet"

Reviewer's Comment:

L 221: "well" occurs twice in sentence. Delete first occurrence.

Answer

Deleted first occurrence of "well"

Reviewer's Comment:

Q: [R3] of Khiat et al in the Replies to Reviewers does not appear in the paper itself. Please, check whether this is indeed what you intend.

Answer:

[R3] was used just in reply to reviewer comments. We do not use the reference in our article and supporting information.

That's all